# Comparison of the Trachea in Normocephalic versus Brachycephalic Cats on the Basis of CT-Derived Measurements

**DOI:** 10.3390/vetsci10100602

**Published:** 2023-10-03

**Authors:** Anna Brunner, Julius Underberg, Jeannette Zimmermann, Simona Vincenti

**Affiliations:** 1Division of Small Animal Surgery, Department of Clinical Veterinary Medicine, Vetsuisse Faculty, University of Bern, 3012 Bern, Switzerland; annabrunnerand@gmail.com (A.B.); j@sofe.de (J.Z.); 2Division of Small Animal Radiology, Department of Clinical Veterinary Medicine, Vetsuisse Faculty, University of Bern, 3012 Bern, Switzerland; julius.underberg@unibe.ch

**Keywords:** brachycephalic cat, CT, feline trachea, normocephalic cat, tracheostomy

## Abstract

**Simple Summary:**

Brachycephalic animals show a shorter shape of the skull than typical for its species, and it is accompanied by multiple development problems. Tracheal hypoplasia, an underdevelopment of the trachea, is a major concern in brachycephalic dogs, but there is no consensus for the trachea in brachycephalic cats. This study aimed to compare tracheal length and diameter between normo- and brachycephalic cats using computed tomography image measurements and evaluate their usefulness in tracheostomy planning. The results of the study indicated no sign of tracheal hypoplasia in brachycephalic cats. Both brachycephalic and normocephalic cats have the same tracheal shape that starts as a circle, then turns oval, and finishes again as a circle. The location between the 4th and 5th cervical vertebrae seems to be the best place to perform a tracheostomy in cats due to its round shape and easily accessible anatomical location.

**Abstract:**

Tracheal hypoplasia is a major concern in brachycephalic dogs, but there is no consensus for the trachea in brachycephalic cats. We aimed to compare tracheal length and diameter between normo- and brachycephalic cats using computed tomography (CT) image measurements and evaluate their usefulness in tracheostomy planning. A total of 15 normocephalic and 14 brachycephalic cats were included in the study. Tracheas of normocephalic cats were significantly longer compared with brachycephalic cats. No difference was detected in tracheal diameter between normocephalic and brachycephalic cats. Both groups had a lateral diameter significantly larger than the dorsoventral diameter at the level of the cranial end of the manubrium sterni and at the level of the second rib. Normocephalic and brachycephalic cats’ tracheas have the same dorsoventral flattening at the level of the cranial end of the manubrium sterni and at the level of the second rib. The location between the 4th and 5th cervical vertebrae seems the best place to perform a tracheostomy in cats due to its round shape and easily accessible anatomical location. No sign of tracheal hypoplasia in brachycephalic cats was detected. Finally, 7 mm appears to be an adequate diameter for the tracheal tubes used to perform feline tracheostomies.

## 1. Introduction

The feline trachea consists of an elastic membranous tube that extends from the cricoid cartilage of the larynx to the base of the heart, where it terminates at the carina before bifurcating into the principal bronchi. It is made up of a series of approximately 35 C-shaped parallel hyaline cartilage rings that are connected dorsally by the trachealis muscle [1].

Feline tracheal hypoplasia, an underdevelopment of the trachea, has been rarely reported and was described in association with mucopolysaccharidosis in only one case report [2]. More frequently observed in cats is dynamic tracheal collapse, which can result from congenital malformation of the cartilage [3,4], upper airway obstruction or neoplasia in the wall or lumen, or as a complication associated with permanent tracheostomy [5,6,7,8,9,10].

Canine tracheal hypoplasia is commonly observed in brachycephalic dogs [11] and can be seen as a component of the brachycephalic syndrome [12]. Although this condition can be clinically silent, it may exacerbate cardiovascular or respiratory disease [12]. The most commonly performed tracheal surgeries in brachycephalic dogs are temporary or permanent tracheostomies, which are associated with a poor prognosis [13,14] due to the significantly smaller trachea and slightly lower mean tracheal thoracic inlet diameter of brachycephalic versus non-brachycephalic dogs [15], frequent concurrence of laryngeal collapse [14], regurgitation and pre-existing gastrointestinal problems [13], and simultaneous operation of the brachycephalic syndrome [13]. These same procedures are associated with a fair-to-good prognosis in normocephalic dogs [16,17].

Permanent tracheostomies are indicated in cats in the case of obstructive masses in the region of the larynx/pharynx, trauma to the upper airways, or stricture. They are rarely performed on cats and seem to have higher complication rates than in dogs or large animals [18,19,20].

A well-described reason for the poor prognosis of cats undergoing tracheostomies is exudative occlusion on the tracheostomy site resulting from the abundant and thick secretion originating from the tracheal epithelium [21]. In temporary tracheostomies or in the first post-operative phase of permanent tracheostomies, recourse to a tracheal tube is often necessary. However, due to its small size, the feline trachea may not allow for adequate ventilation around the tube in case of occlusion [19,21]. In addition to tracheal obstruction due to massive secretion, other reported complications include stricture, focal tracheal collapse, pneumonia, swelling at the surgery site, and pulmonary oedema [8,22,23,24], resulting in a high associated mortality rate [1,22,23,24]. There is little doubt, in the authors’ opinion, that improving the safety of tracheostomies in cats requires a better knowledge of the tracheal anatomy.

Currently, the tracheostomy technique used in cats is identical to the one developed for dogs, which have a trachea that is circumferential for most of its length [25,26]. Furthermore, to the best of the authors’ knowledge, CT-derived measurements of tracheal dimensions in normocephalic versus brachycephalic cats have not been previously reported. The only published measurements of tracheal diameters (Persian versus Domestic Shorthair cats) were taken from lateral radiographs at the thoracic inlet and at the level of the third rib [27] and from CT images but only in normocephalic cats [28].

The lack of available data might be the reason why clinicians refrain from performing tracheostomies on brachycephalic cats. In order to provide new knowledge about the anatomy of the trachea in brachycephalic cats and possibly support the surgeon during feline tracheal surgical planning, the aim of the study was to compare the variations in length and diameter of tracheae in normocephalic versus brachycephalic cats and to evaluate their usefulness on planning a tracheostomy in cats.

## 2. Materials and Methods

### 2.1. Study Design and Inclusion Criteria

This retrospective study included CT images from 15 normocephalic and 14 brachycephalic client-owned cats (*n* = 29). The CT images existed in-house as part of the data compiled by the Small Animal Hospital of the Vetsuisse Faculty at the University of Bern. The inclusion criteria for the normocephalic group comprised normocephalic conformation, a body weight of more than 2 kg, a minimum age of 12 months, no evidence of acute or chronic airway disease, a full medical record, a standardised neck and thoracic CT scan, and not having the endotracheal cuff inflated at the measured points. The same criteria were used for the brachycephalic group, except that only brachycephalic breeds were included (Persian, British Shorthair, Exotic Shorthair, Scottish Fold, and Himalayan).

A 16-slice CT unit (Philips Brilliance 16, Philips Medical Systems Nederland BV, Best, the Netherlands) was used for all patients. Cats were positioned in sternal recumbency. Mechanical hyperventilation was applied before scanning to suppress respiratory drive, and a mild positive pressure of 10 cm of water was maintained during image acquisition to obtain images in inspiration. The scanning direction was caudal to cranial. Technical parameters were 120 kV, 150 mAs, a collimator pitch of 0.938, a 1 mm slice, a rotation time of 1 s, a field of view of 250 mm, and a 512 × 512 matrix. Post-contrast scans were acquired 20 s after the end of manual administration of an iodinated contrast medium (Accupaque 300 mg I/mL, GE Healthcare AG, Glattbrugg, Switzerland) through a cephalic vein catheter at a dose of 2 mL/kg bodyweight. Images were reconstructed using standard and high-spatial-frequency algorithms and viewed on an external workstation using a commercially available software (DeepUnity Diagnost, DH HealthCare GmbH, Bonn, Germany).

The anaesthesia protocol was standardised for all subjects according to their ASA score: ASA I subjects were anaesthetised using intravenous medetomidine (0.01–0.08 mg/kg; Domitor ad us. vet), butorphanol (0.2–0.3 mg/kg; Morphasol-10 ad us. vet), and propofol (to effect; Propofol-Lipuro); ASA II-III subjects received intravenous methadone (0.2 mg/kg), ketamine (1–2 mg/kg; Ketanarkon 100 ad us. vet), midazolam (0.1–0.2 mg/kg; Dormicum), and propofol to effect. After endotracheal intubation, anaesthesia was maintained with isoflurane (titrated to effect; Isoflo ad us. vet) and oxygen (60–100%).

### 2.2. Tracheal Measurements

Images were reformatted to the sagittal plane with a slice thickness of 1 mm and an overlap of one-third using soft tissue and bone algorithms. The images were exported to a workstation for review in pulmonary (−600 level/1600 width) and soft tissue (160 level/600 width) windows. Pre- and postcontrast images were used, depending on which was considered most adequate by the decision of the reviewers upon agreement. The transverse plane was adapted at mentioned locations to be perpendicular to the trachea, and lateral and dorsoventral inner diameters were measured at five points (Figure 1 and Figure 2).

Measure 1 (M1): 1 cm caudal to the cricoid cartilage;

Measure 2 (M2): at the level of the C4–C5 intervertebral disc space;

Measure 3 (M3): at the cranial end of the manubrium sterni;

Measure 4 (M4): at the level of the second rib;

Measure 5 (M5): 1 cm cranial to the carina.

Length was measured with the multiplanar reconstruction curve tool from 1 cm caudal to the cricoid cartilage to 1 cm cranial to the carina as a curvilinear line along the central lumen of the trachea. All measurements were once performed by the first author (A.B.) and three separate times at different time points by the second author (J.U.).

### 2.3. Statistical Analyses

A mean value of the four measured values was calculated and used for the statistical analyses. All statistical analyses were performed using MedCalc (MedCalc Software 22.013), a commercially available software. Significance was set at *p* < 0.05. Descriptive statistics (frequency and summary statistics tables) were calculated for clinical analyses. A Fisher’s exact test was used to assess differences between groups for categorical clinical data (sex). As some data were normally distributed but others were not (and could not be transformed to normality), we used a two-tailed *t*-test for normally distributed data (age, weight, length, and diameter of the trachea) and a Mann–Whitney *U* test for not normally distributed data. The power of this study was assessed using a post hoc power calculation based on existing dog-related data [25]. Using an alpha value of 0.05 and a study sample of 15 normocephalic subjects and 14 brachycephalic subjects, a power of 100% was obtained.

## 3. Results

### 3.1. Cohort Characteristics

A total of 29 cats were included in the study: 15 in the normocephalic group and 14 in the brachycephalic group. Of the 15 normocephalic cats, 9 were castrated male and five were spayed female. The mean age was 10.491 years (range 1.33–16.91). Breeds included Domestic Shorthair (*n* = 12), Oriental Shorthair (*n* = 1), Bengal (*n* = 1), and Egyptian Mau (*n* = 1). The mean body weight was 4.58 kg (range 2.45–8). Indications for CT imaging included multiple fractures of the mandibula/maxilla (*n* = 2), feline hippocampal necrosis (*n* = 1), skull fracture (*n* = 1), chronic otitis (*n* = 2), chronic right front limb lameness (chronic flexor enthesopathy) (*n* = 1), and tumour staging (*n* = 8). The diagnosis and localisation of the eight tumours included fibrosarcoma of the left hip (*n* = 1), mandibular squamous cell carcinoma (*n* = 2), subcutaneous soft tissue sarcoma in the dorsal region of the neck of 2.0 cm diameter (*n* = 1), mediastinal lymphoma with no mass effect on the trachea (*n* = 2), splenic mast cell tumour (*n* = 1), and squamous cell carcinoma of the middle ear (*n* = 1).

Of the 14 brachycephalic cats, 9 were castrated male and 5 were spayed female. The mean age was 10.304 (range 0.920 to 17.750). Breeds comprised Persian (*n* = 10) and British Shorthair (*n* = 4). Indications for CT imaging included retrobulbar abscess (*n* = 2), vertebral disc extrusion/protrusion (*n* = 2), bilateral hydrocephalus (*n* = 1), arachnoid cyst (*n* = 1), full body screening after severe trauma (*n* = 1), mandibular fracture (*n* = 2), and tumour staging (*n* = 5). The diagnosis and localisation of the five tumours were retrobulbar squamous cell carcinoma (*n* = 1), mandibular squamous cell carcinoma (*n* = 2), soft tissue sarcoma of the shoulder (*n* = 1), and osteosarcoma of the scapula (*n* = 1).

The mean body weight of the brachycephalic cats was 3.625 kg (range 2.600–6.000). The brachycephalic group had a significantly lower body weight (*p* = 0.050; Mann–Whitney *U* test) than the normocephalic group. We found no difference in the mean body weight/trachea length ratio between the two groups (normocephalic 0.034; brachycephalic 0.031). No difference was found in terms of age and sex distribution between normocephalic and brachycephalic cats.

### 3.2. Trachea

The median and range of all measurement points of the trachea in normocephalic versus brachycephalic cats are summarised in Table 1. The trachea of normocephalic cats was significantly longer (mean 135.747 mm, +/−12.785 mm) than that of brachycephalic cats (mean 115.085 mm, +/−8.224 mm) (*p* = 0.0001; *t*-test). Comparisons between the normocephalic and brachycephalic groups revealed no difference in the lateral and dorsoventral diameters of the trachea or in the ML/DV diameter ratios at any measurement point.

In the normocephalic group, the lateral diameter of the trachea was observed to be significantly longer than the dorsoventral diameter at the level of the cranial end of the manubrium sterni (*p* = 0.035 at M3 in Table 2; Mann–Whitney *U* test) and at the level of the second rib (*p* = 0.005 at M4 in Table 2; Mann–Whitney *U* test). In the brachycephalic group, a significantly longer lateral diameter compared to the dorsoventral diameter was observed at the same levels (*p* = 0.031 at M3 in Table 3, Mann–Whitney *U* test, and *p* = 0.035 at M4 in Table 3, Mann–Whitney *U* test). At the other measurement points, there was no difference between lateral and dorsoventral diameters in either group.

## 4. Discussion

The literature under review contains very limited data on the comparison of tracheal measurements between normocephalic and brachycephalic cats [27]. The results of our study show no difference in the lateral and dorsoventral diameters of the trachea of normocephalic versus brachycephalic cats, whatever the measurement point. Those results are in accordance with the study of Hammond et al. [27] where no difference in the absolute size of the trachea of Persian versus Domestic Shorthair cats was found using radiographic measurements. Evidence suggests that, unlike what has been observed in dogs, no sign of tracheal hypoplasia can be found in brachycephalic cats [12] using radiography or CT imaging.

For both normocephalic and brachycephalic groups, the feline trachea is elliptical in shape at the level of the cranial end of the manubrium sterni and at the level of the second rib, suggesting that it is dorsally and ventrally flattened at these points. In the sample under study here, the diameter of the trachea is circumferential in the cervical portion then progressively elliptical at the level of the thoracic inlet before becoming again almost circumferential at the carina. The flattening of the trachea at the level of the thoracic inlet may be explained by the curvature and the surrounding structures such as the oesophagus, bones, and muscles. A similar observation was made previously in dogs [29], and our results are in accordance with a previous study about the tracheal measurement of normocephalic cats on CT images, as they also found an elliptical shape with dorsoventral flattening at the level of the thoracic inlet [28].

The flattening of the trachea at the junction of the thoracic inlet may be a predisposing factor for primary tracheal collapse in this area, a phenomenon infrequently described in cats [30]. Knowledge of a dorsally and ventrally flattened trachea at the level of the thoracic inlet could be useful information in the event of tracheal surgery. Indeed, a post-surgical tracheal collapse might be more likely in this area compared to the cervical portion where the trachea is almost circumferential. There are three locations where the trachea is rounder: 1 cm caudal to the cricoid cartilage (M1), between C4 and C5 (M2), and 1 cm cranial to the carina (M5). For reasons of accessibility, in the authors’ opinion, the second of the three locations would be recommended for performing a tracheostomy.

Our study reveals a difference in the average trachea length between normocephalic and brachycephalic cats. Such a finding is most likely due to the significantly larger body weight of normocephalic versus brachycephalic cats because there was no difference in body weight/trachea length ratio between the two groups.

In the authors’ opinion, the data produced in this study could be helpful in establishing a database of tracheal lengths and diameters in normocephalic and brachycephalic cats using CT imaging and in determining the grade of narrowing and flattening of the trachea in cats with suspected tracheal disease.

Moreover, the choice of an adequately sized tracheal tube in temporary tracheostomies or in the first post-operative phase of permanent tracheostomies is crucial in cats as it allows for the proper ventilation required by an increased risk of obstruction due to mucus secretion [22]. Our study indicates that the tracheal diameter varies between 7 mm and 8 mm, in both normocephalic and brachycephalic cats, except at the level of the cranial end of the manubrium sterni and at the level of the second rib. From our results, a tracheal tube of 7 mm in diameter would appear to be the best suited for tracheostomies if no measurement can be made before the intervention.

Finally, it should be noted that our study has some limitations, including the retrospective nature of the analysis and the use of hospital patients. Even if the cats were selected based on an absence of airway or neck disease, the likelihood of a biased sample cannot be totally excluded. Furthermore, no evaluation for dynamic tracheal changes was performed; therefore, we cannot completely exclude a dynamic collapse of the trachea. The images were obtained in inspiration, and an expiratory collapse cannot therefore be excluded. In addition, although no measurement was performed at the level of the endotracheal cuff, a minimal change in the diameter induced by the presence of the cuff nearby cannot be ruled out. Finally, we did not correlate the tracheal length with the thorax size or the neck size; therefore, the effect of a short thorax or a short neck on the length in the brachycephalic cats cannot be excluded. However, we correlated the body weight with the tracheal length, and we found a similar ratio in both groups.

## 5. Conclusions

Conversely to what has been observed in dogs, CT imaging reveals no difference in the shape of the trachea in normocephalic versus brachycephalic cats. The authors’ assumption is that the difference observed in tracheal length between the two groups results from the considerably lower body weight associated with the smaller size of brachycephalic cats.

The results of our study show no evidence of an anatomical predisposition to a worse prognosis associated with tracheal surgery for brachycephalic cats. Nevertheless, further clinical and biomechanical studies are required in order to develop a technique adapted to feline subjects. As has been pointed out, the tracheotomy technique used on dogs cannot be directly transferred to cats because of the different anatomy of canine versus feline tracheae.

It appears that the location between the 4th and 5th cervical vertebrae is the best place to perform a tracheostomy in cats because of its round shape and its easily accessible anatomical location. Finally, our results suggest that 7 mm is the most suitable diameter for the tracheal tube used when performing feline tracheostomies.

## Figures and Tables

**Figure 1 vetsci-10-00602-f001:**
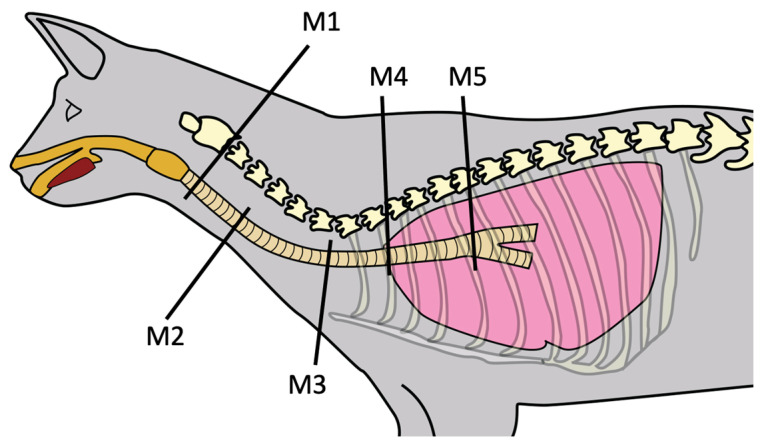
Schematic representation of the feline trachea, lateral view, showing areas where the five different tracheal measurements (M1, M2, M3, M4, and M5) were performed.

**Figure 2 vetsci-10-00602-f002:**
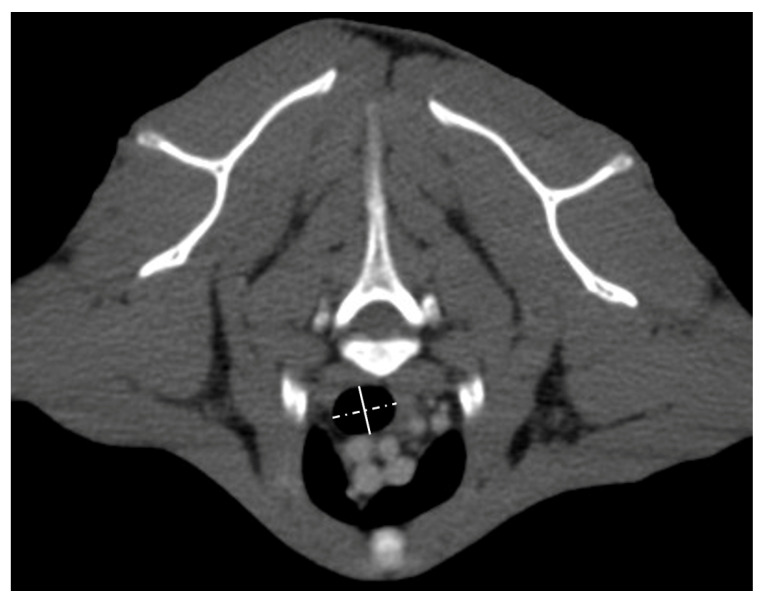
Representative contrast-enhanced transverse image in soft tissue window at the level of the cranial end of the manubrium sterni with the measurements of the lateral (dotted line) and dorsoventral (closed line) inner diameter of the trachea.

**Table 1 vetsci-10-00602-t001:** Comparison of tracheal measurements in normocephalic versus brachycephalic cats.

Variable		Normocephalic Cats (*n* = 15)	Brachycephalic Cats (*n* = 14)	*p*-Value
Length (mm)		135.747 (+/−12.785 mm)	115.085 (+/−8.224)	0.0001
M1	ML (mm)	7.425 (5.725–11.575)	7.500 (6.475–9.000)	0.827
	DV (mm)	7.575 (6.525–11.475)	7.375 (6.050–8.000)	0.183
	Ratio ML/DV	1.000 (0.814–1.116)	1.062 (0.863–1.210)	0.052
M2	ML (mm)	8.200 (5.425–9.650)	8.000 (6.300–9.700)	0.984
	DV (mm)	7.175 (6.475–9.750)	7.550 (5.100–8.550)	0.570
	Ratio ML/DV	1.096 (0.746–1.306)	1.087 (0.965–1.382)	0.704
M3	ML (mm)	8.550 (6.150–9.625)	8.000 (5.975–9.775)	0.153
	DV (mm)	7.650 (5.400–9.500)	7.487 (5.250–8.625)	0.325
	Ratio ML/DV	1.120 (0.992–1.339)	1.145 (1.020–1.306)	0.654
M4	ML (mm)	8.200 (5.975–9.000)	7.213 (5.600–10.070)	0.079
	DV (mm)	7.200 (5.150–8.475)	6.512 (5.075–8.800)	0.381
	Ratio ML/DV	1.124 (1.026–1.432)	1.141 (1.038–1.336)	0.513
M5	ML (mm)	7.500 (4.875–8.175)	6.700 (5.475–8.600)	0.085
	DV (mm)	7.425 (5.575–8.550)	6.987 (5.350–8.475)	0.507
	Ratio ML/DV	1.023 (0.874–1.218)	1.003 (0.859–1.086)	0.425

Values are presented as mean (+/−standard deviation) for normally distributed data and as median (min–max) for not normally distributed data; M1, measure 1; M2, measure 2; M3, measure 3; M4, measure 4; M5, measure 5; ML, lateral diameter; DV, dorsoventral diameter; *p*-value, between-group comparison using the Mann–Whitney *U* test (not normally distributed) or *t*-test (normally distributed).

**Table 2 vetsci-10-00602-t002:** Tracheal measurements in normocephalic cats.

Measurement Point	ML	DV	Ratio ML/DV	*p*-Value
M1	7.425 (5.725–11.157)	7.575 (6.525–11.475)	1.000 (0.814–1.116)	0.885
M2	8.200 (5.425–9.650)	7.175 (6.475–9.750)	1.096 (0.746–1.306)	0.144
M3	8.550 (6.150–9.625)	7.650 (5.400–9.500)	1.120 (0.992–1.339)	0.035
M4	8.200 (5.975–9.000)	7.200 (5.150–8.475)	1.124 (1.026–1.432)	0.005
M5	7.500 (4.875–8.175)	7.425(5.575–8.550)	1.023 (0.874–1.218)	0.724

Values are presented as mean (+/−standard deviation) for normally distributed data and as median (min–max) for not normally distributed data; M1, measure 1; M2, measure 2; M3, measure 3; M4, measure 4; M5, measure 5; ML, lateral diameter; DV, dorsoventral diameter; *p*-value, between-group comparison using the Mann–Whitney *U* test (not normally distributed) or *t*-test (normally distributed).

**Table 3 vetsci-10-00602-t003:** Tracheal measurements in brachycephalic cats.

Measurement Point	ML	DV	Ratio ML/DV	*p*-Value
M1	7.500 (6.475–9.000)	7.375 (6.050–8.000)	1.062 (0.863–1.210)	0.136
M2	8.000 (6.300–9.700)	7.550 (5.100–8.550)	1.087 (0.965–1.382)	0.113
M3	8.100 (6.325–9.775)	7.487 (5.250–8.625)	1.145 (1.020–1.306)	0.031
M4	7.350 (5.600–10.075)	6.512 (5.075–8.675)	1.141 (1.038–1.336)	0.035
M5	6.700 (5.475–8.600)	6.987 (5.350–8.475)	1.003 (0.859–1.086)	0.725

Values are presented as mean (+/−standard deviation) for normally distributed data and as median (min-max) for not normally distributed data; M1, measure 1; M2, measure 2; M3, measure 3; M4, measure 4; M5, measure 5; ML, lateral diameter; DV, dorsoventral diameter; *p*-value, between-group comparison using the Mann–Whitney *U* test (not normally distributed) or *t*-test (normally distributed).

## Data Availability

The data that support the findings of this study are available from the corresponding author, S.V., upon reasonable request.

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
