# Peer review of "Comparison of the Trachea in Normocephalic versus Brachycephalic Cats on the Basis of CT-Derived Measurements"

_vetsci, 2023, doi:10.3390/vetsci10100602_

Round 1

Reviewer 1 Report

This is the paper to compare tracheal diameters in brachycephalic cats and normocephalic cats using CT, which is interesting because of the increasing number of reports of BOAS in cats. However, there are several points in this paper that need to be reconsidered.

Title, Simple Summary, Introduction, Discussion

The main objective is to determine the optimal site and impact for tracheostomy in cats, but this conclusion cannot lead the results of this study. The site of the tracheostomy is only presumed in the discussion. To include it in the title or main objective is an overspeculation and should be reconsidered.

Line58-64  Since permanent tracheostomy in cats is itself considered a very difficult technique, it makes no significant sense to discuss the impact of brachycephalic breed.

Line95-96  You state that there is no airway disease, but do you evaluate for dynamic tracheal changes? This study focused on the diameter of the trachea and it should be confirmed that there is no dynamic collapse affecting the tracheal diameter

Line137  Why did you take one and three measurements by two authors? Why not two times each?

Line182-183 Please illustrate where mediolateral and dorsoventral diameters of trachea are in Figure1 or any other figure.

Line215-217,259-260 Can you sure that there is no tracheal hypoplasia in brachycephalic breed cats? The cats included in this study were limited to a few breeds of the brachycephalic breed. Furthermore, can the diameter of the trachea alone be used to diagnose hypoplasia? In dogs, tracheal diameter measurement against the thoracic inlet is used, therefore, shouldn't the effect of body size, such as a narrow thorax, be taken into account?

Line179-181, 240-243 In relation to the above, the length of the trachea should not be considered simply as an effect of brachycephalic breed, but should also take into account weight and narrow thorax.

Reviewer 2 Report

The review of a paper entitled "Comparison of the trachea in normocephalic versus brachycephalic cats on the basis of CT-derived measurements and impact on tracheostomy location".

Dear Authors,

I read your paper with a great interest. In my opinion it is a valuable and interesting work. Nonetheless, some minor revisions should be made to improve the paper, as follows:

- Introduction section: In the whole section the information about cats are mixed with information about dogs, sometimes leading to confusion; e.g. in lines 39-45 you first describe the anatomy of feline trachea to mention canine tracheal hypoplasia next; in my opinion, it would be clearer if line 43 was moved to another paragraph together with the next lines regarding feline hypoplasia; similarly, in lines 50-forward, you describe surgeries performed in dogs after describing tracheal hypoplasia in cats; consider moving that part to another paragraph; please revise the whole Introduction section to make it more clear for a reader

- lines 43 and further: in my opinion it would be valuable to shortly describe what is tracheal hypoplasia before mentioning its frequency in dogs and cats also because in the Discussion you mention that signs of hypoplasia are not observed in cats (line 216)

- lines 46-47: although you mention that feline tracheal hypoplasia is rarely reported, you cite only one paper describing a case report; to be honest, I have not found any other report on that disease in cats; therefore, I think it would be valuable to emphasize that there is only one report; nonetheless, if you found any other reports of that disease in cats, you should mention them here

- lines 76 and further: in my opinion adding a short list of indications for tracheostomy in cats (and maybe dogs) would be valuable to emphasise the value of your work

- lines 102-103: please explain if the mechanical hyperventilation and maintained positive pressure could have had an impact on the obtained measurements and further on the recommendation of the use of 7mm tube; if it could, please add that shortcoming to the limitation paragraph

- for the whole Results section: please provide the type of statistical test used for each result (preferably next to p-value); moreover, as you mention that some of the data were parametrical while others were not, please present the parametrical data as mean +/- SD and leave non-parametrical data as median (range); the same rule should be applied in tables;

- line 165: please correct the typo in "spayed"

- lines 166-167: in Materials and methods section you mention 5 brachycephalic breeds (line 99), while only two of them were used; please consider unifying the descriptions

- line 236: explain what you mean by "larger size"? I assume that larger body weight but it is unclear; moreover I think it would be valuable to check the correlation between the body weight and tracheal length as an addition to performed measurements, as you report that there was a significant difference between BW between two groups and hypothesise on the reasons of shorter trachea in the brachycephalic group; in my opinion, a shorter trachea may be also resulting from a shorter neck; please consider taking other measurements, e.g. neck length or vertebrae length to compare both groups and look for correlations; I do not consider the latter essential, but in my opinion, it would improve the results and possible discussion

- line 237: please explain "shorter thoracic size" - do you mean length, width, hight, volume? Did you measure the thoracic size or is it a literature information (if so, please provide appropriate reference). If it is only your assumption, please indicate that

- lines 262-263: please see above regarding the factors influencing tracheal length

Reviewer 3 Report

Dear Authors

I reviewed the manuscript entitled "Comparison of the trachea in normocephalic versus brachycephalic cats on the basis of CT-derived measurements and impact on tracheostomy location". The manuscript is well written and the topic is interesting, since little is known about the comparison of tracheal diameter between brachycephalic and normocephalic cats. Nonetheless I have some important concerns on materials and methods section (see specific comments). Therefore I recommend major revision.

Specific comments: 

line 82: please add a citation of a recent paper entitled "CT measurements of tracheal diameter and length in normocephalic cats" of  the same group of Authors (Zimmermann et al, Journal of Feline Medicine and Surgery, Volume 25Issue 3March 2023)

line 103: "Mechanical hyperventilation was applied before scanning to suppress respiratory drive and a mild positive pressure of 10 cm of water was maintained during image acquisition". Therefore measurements obtained are in inspiration (see Leonard et al, "Changes in tracheal dimensions during inspiration and expiration in healthy dogs as detected via computed tomography", AJVR, 2009). This should be mentioned throughout the manuscript and addressed in the discussion.

lines 120-122: Sentences are unclear. Which window width/level was used for measurements?

liner 125-126: please add a more thorough description of tracheal measurements. Which tracheal diameter did the Authors consider? Internal or external? (see Kaye et al "Computed tomographic, radiographic, and endoscopic tracheal dimensions in english bulldogs with grade 1 clinical signs of brachycephalic airway syndrome" Vet Radiology and Ultrasound, 2015)

lines 137-138: Sentence is unclear. Were mean values obtained? Was inter-observer reliability calculated?

line 166: Please standardize the number of digits after the decimal point throughout the manuscript.

Results:

please, change "medio-lateral" with "lateral" diameter throughout the manuscript. 

Discussion

The measurements obtained in this study are only in inspiration. Therefore the variation in tracheal dimensions during respiration was not assessed. This may be another limitation of the study, please add a comment about this issue.

Figures: please add more figures of the different measurement points on CT images. This may be of interest to the reader.

Reviewer 4 Report

Interesting work. Well described and written. 

Few comments to make: 

  • Did you check that the anesthetics used in this project do not affect the bronchial/tracheal movement, like butorphanol can do? If so, it would be nice to have a small comment about that. 

  • About the results, the difference in tracheal length is really interesting but I feel you didn't work on this point. My recommendations: 

  • Check which part of the trachea is longer in normal cats. 

  • Make a compensation/ratio between the tracheal length and the body weight and see if there's any difference between groups. Do you have the total body length? It would be even better than the weight for correcting with the trachea. If not, maybe you have the whole thorax imaged. This could be another way of correcting the tracheal length. 

  • For the tracheal collapse that you refer to in the discussion, in my opinion it should be necessary to have a dynamic study of the trachea to check this. Static studies are not the most accurate for analyzing dynamic problems. By chance, do you have respiratory gating in your scan? If so, why don't you compare inspiration Vs expiration tracheal sizes? 

  • I recommend including an image of how you measure the tracheal diameter for better understating the protocol. And a 3D render of a normal and brachiocephalic trachea would be great too for confirming the similarity in shapes.

Round 2

Reviewer 3 Report

Dear  Authors,

I reviewed the manuscript. My concerns have been addressed accordingly. Therefore I recommend publication.